# Infant Milk Formula Enriched in Dairy Cream Brings Its Digestibility Closer to Human Milk and Supports Intestinal Health in Pre-Clinical Studies

**DOI:** 10.3390/nu16183065

**Published:** 2024-09-11

**Authors:** Alina Kondrashina, Gianfranco Mamone, Linda Giblin, Jonathan A. Lane

**Affiliations:** 1Health and Happiness (H&H) Group, H&H Research, Global Research and Technology Centre, Fermoy, P61 K202 Co. Cork, Ireland; 2Institute of Food Science, National Research Council, 83100 Avellino, Italy; 3Teagasc Food Research Centre, Moorepark, Fermoy, P61 P302 Co. Cork, Ireland

**Keywords:** infant formula, infant nutrition, milk fat globule, dairy lipids, cream

## Abstract

Human breast milk (HBM) is the “gold standard” for infant nutrition. When breast milk is insufficient or unavailable, infant milk formula (IMF) can provide a safe and nutritious alternative. However, IMFs differ considerably from HBM in composition and health function. We compared the digestibility and potential health functions of IMF containing low cream (LC-) or high cream (HC-) with pooled HBM. After simulated infant digestion of these samples, the bioavailability of key nutrients and immunomodulatory activities were determined via cell-based *in vitro* assays. A *Caenorhabditis elegans* leaky gut model was established to investigate cream effects on gut health. Distinct differences were observed in peptide diversity and sequences released from HC-IMF compared with LC-IMF during simulated digestion (*p* < 0.05). Higher levels of free fatty acids were absorbed through 21-day differentiated Caco-2/HT-29MTX monolayers from HC-IMF, compared with LC-IMF and HBM (*p* < 0.05). Furthermore, the immune-modulating properties of HC-IMF appeared to be more similar to HBM than LC-IMF, as observed by comparable secretion of cytokines IL-10 and IL-1β from THP-1 macrophages (*p* > 0.05). HC-IMF also supported intestinal recovery in *C. elegans* following distortion versus LC-IMF (*p* < 0.05). These observations suggest that cream as a lipid source in IMF may provide added nutritional and functional benefits more aligned with HBM.

## 1. Introduction

Human breast milk (HBM) is widely regarded as the best dietary choice for supporting growth and development in early infancy. However, despite the nutritional value of HBM, it is estimated that only 38% of infants globally are exclusively breastfed during the first 6 months of life for a variety of medical or personal reasons [1]. Infant milk formulations (IMFs), commonly produced from cow’s milk with additional supplementation of essential fatty acids from vegetable oils, aim to mimic the composition of HBM. Despite strict US Food and Drug Administration guidelines on the manufacturing of IMF, its functionality can be markedly different from HBM due to different sources, structures, and ratios of individual ingredients, as well as the impact of IMF processing [2]. Thus, it is crucial to continually compare IMF with HBM to improve its composition and maximize its nutritional and functional value. 

HBM contains a bioactive mix of components, such as peptides, oligosaccharides, immunoglobulins, hormones, growth factors, cytokines, chemokines, and lipids, which vary in composition to match the nutritional needs of the infant at each stage of development [3,4]. Carbohydrates, predominantly in the form of lactose, are the most abundant macronutrient in HBM, while lipids are the second most abundant, accounting for approximately 40–50% of the total energy content, depending on the stage of lactation [3,5]. There are approximately 200 different fatty acids in HBM [6]. As with all mammalian species, milk fats in HBM are packaged and delivered in milk fat globules (MFGs) to allow them to remain emulsified in the aqueous phase, while the structure of MFG is said to aid in milk digestibility [7]. Triglycerides and essential fatty acids found at the core of the MFG make up approximately 95–98% of the total fatty acid content in HBM [3,8,9]. Palmitic acid (PA) is the most abundant saturated fatty acid in HBM, accounting for approximately 17–25% of all fatty acids [10]. Furthermore, it has been reported that around 75% of PA in HBM is esterified in the sn-2 position (β-palmitate). This is an important feature of HBM since β-palmitate remains unhydrolyzed during the digestion of triglycerides and can be more easily absorbed, along with essential nutrients such as calcium [9,10], therefore adding to the digestibility and functionality of HBM. 

MFGs are surrounded by a delicate bioactive tri-layer membrane known as milk fat globule membranes (MFGMs), which are produced by lactocytes in the mammary gland. MFGMs are an abundant source of additional nutrients or bioactives, such as phospholipids, glycolipids, glycoproteins, and carbohydrates [8]. For example, MFGM is particularly rich in phospholipids, such as phosphatidylcholine, and sphingolipids, such as sphingomyelin and gangliosides, as well as cholesterol [8]. Thus far, a total of 191 proteins have been identified solely from HBM MFGM [8,11]. The specific combination and structure of nutrients in HBM, particularly those found in the MFG/MFGM, play a vital role in neurological development, establishing the gut microbiome and the immune system, and maintaining the intestinal barrier function in breastfed infants [3,8]. Indeed, extensive pre-clinical and clinical research has highlighted the potential health benefits of several MFG components, such as cholesterol and sphingomyelin, particularly on neurodevelopmental outcomes in infants [8,12]. 

By contrast, dairy IMFs generally contain little or no naturally occurring MFG/MFGM due to the frequent use of low-fat skimmed milk powder as a base, whereby most of the dairy fat portion has been removed during processing [7,8,13]. Supplementation with vegetable oils offers a cost-effective means of incorporating essential fats into a formula to compensate for the lack of MFGM. However, while this approach may provide many essential lipids with the desired proportions of mono- and polyunsaturated fats, the overall fatty acid profile of the resulting IMF may be markedly different from HBM in composition and functionality. For example, it is estimated that only approximately 20–30% of PA in vegetable oils is β-palmitate, compared to 75% in HBM. Instead, the majority of PA in vegetable oils is esterified in the sn-1 and sn-3 positions [9,10,14]. As previously stated, this key difference in the structure of PA can decrease the digestibility and absorption of lipids and other vital nutrients in IMF supplemented with vegetable oils [9], compared with HBM. Esterification of PA at sn-1 and sn-3 can result in an accumulation of solidified PA molecules in the intestine, which form insoluble complexes with essential dietary minerals, such as calcium, leading to hard stools, constipation, and poor absorption of nutrients [10,15]. Furthermore, supplementation of IMF with vegetable oil cannot replicate peptides and additional nutrients contained in MFGM. Thus, there is a need to improve the fatty acid profile, functionality, and overall nutrient content of IMF using alternatives to vegetable oil sources of lipids. 

One method of doing this is to supplement IMF with commercially available MFGM since many features of MFGM are highly conserved between mammalian species [9]. Manoni and colleagues [9] compared HBM with cow, goat, and yak milk and reported that HBM shared 166 proteins in common with cow’s milk versus only 95 in common with goat’s milk [9]. Likewise, the molecular functionality of MFGM components from HBM was reportedly similar to MFGM from cow’s milk [9]. Bovine MFGM also contains a slightly higher level of PA than HBM (approximately 30% versus 22%). Furthermore, approximately 45% of bovine PA is present as β-palmitate [6], which is nearer to the level found in HBM than the average content in vegetable oils. Indeed, improvements in fatty acid absorption and gut microbiota have previously been reported in infants fed with IMF supplemented with PA, where 43–44% occurred as β-palmitate [10,16], suggesting the potential for enhanced digestibility with bovine MFGM supplementation containing the same proportion of β-palmitate. These findings indicate that a bovine source of MFGM may be optimal for ensuring the closest match to HBM in terms of peptide composition, lipid content, and fatty acid profile, as well as other nutrients associated with MFGM. Results from clinical trials have indicated that supplementation of IMF with bovine MFGM can have potential benefits on cognitive development, gut microbiota, and rate of infection, compared to standard formulas [17,18,19,20,21], though further research is needed to verify these initial findings on a larger scale and in real-world practice. However, despite its promising effects, isolating and processing bovine MFGM can often involve multiple refinement steps, which could have cost implications when considering its use in the IMF industry [22,23,24,25,26]. 

Supplementation of IMF with dairy creams offers an alternative means of providing MFG/MFGM to IMF, with the advantage of requiring minimal processing of the cream. It is estimated that dairy cream (~38% fat) contains up to 200 mg/100 g MFGM [10], which is approximately 13 times greater than the level of MFGM in skimmed milk (~0.5% fat) and 5–6 times greater than the levels found in whole milk (~3.5% fat) [9]. Adding dairy cream to IMF as a fat source would provide MFG/MFGM, thereby improving the overall lipid content and fatty acid profile of IMF and increasing the proportion of functionally beneficial β-palmitate in particular. Furthermore, dairy cream supplementation would also increase the presence of additional nutrients associated with the MFGM, such as carbohydrates and peptides. Consequently, the addition of dairy cream to IMF has the potential to produce formulations that may be more aligned with the “gold-standard” composition of HBM. 

In this study, two IMFs were developed with a new fat blend and varying cream content to achieve a fatty acid profile as close as possible to that of HBM. Subsequently, we investigated how the compositional differences between these two cream-enriched IMFs affected protein and lipid digestibility, bioavailability of key nutrients, and immune and gut health function when compared to pooled HBM. The objective of this study was to determine if using dairy cream as a lipid source in IMF production provided it with additional health benefits.

## 2. Materials and Methods

### 2.1. Samples

Two experimental IMF containing low cream (LC-IMF) and high cream (HC-IMF) were produced on the same day in Moorepark Technology Limited (Teagasc Food Research Centre, Fermoy, Ireland) with double levels of cream in HC-IMF when compared to LC-IMF. Macronutrient and micronutrient content was matched and in accordance with the specifications for stage 1 IMF (0–6 months, protein 10.5 g/100 g (whey:casein 60:40), fat 27 g/100 g, carbohydrates 56.2 g/100 g) [27], except for the cream content. The cream added to both IMFs was sourced from the same batch to avoid seasonal compositional variability. Each formula contained vegetable oil blends, created to yield matching lipid profiles as close as possible to HBM despite different cream content. IMFs were spray-dried, canned, and flushed with nitrogen to ensure optimal storage conditions and a shelf-life of up to 2 years at room temperature. The processing of formulations was closely monitored to ensure that the HC-IMF and the LC-IMF possessed an identical process history. Once opened, IMFs were re-sealed and used within one week. Freeze-dried mature HBM from at least three donors (protein content of 5.7% in reconstituted form) was purchased from Lee BioSolutions (Maryland Heights, MO, USA).

### 2.2. Infant Simulated Gastrointestinal In Vitro Digestion Method

Prior to digestion, 4.5 g of IMF (final product) was reconstituted in 30 mL of water, as per the manufacturer’s instructions. HBM was reconstituted as 133 mg solids in 1 mL purified water to yield the original human milk concentration as per the manufacturer’s protocol. Protein content was 18 ± 2 mg/mL in IMFs and 12 ± 1 mg/mL in HBM, measured by BCA assay (Pierce™ BCA Protein Assay Kit, Thermo Fisher, Dublin, Ireland). Gastrointestinal digestion was performed using the INFOGEST protocol adjusted for the infant’s conditions [28]. The infant protocol included gastric enzymes at activities of 268 U/mL pepsin and 19 U/mL lipase, a gastric pH of 5.3, and time in the gastric phase of 1 h. The intestinal phase was performed with 16 U/mL trypsin activity supplied by pancreatin and 3.1 mM of bovine bile salts at pH 6.6 for 1 h. Prior to digestion, enzymatic assays were performed in-house [29]. Digested samples were collected before the gastric phase (G0), after the gastric phase (G60), and after the intestinal phase (I60). Separate tubes were set up for each digestion collection point to avoid the heterogenicity of samples. Digestion in gastric samples was stopped by adjusting pH to 7 and treatment with a lipase inhibitor orlistat (1 mM, Merck, Carrigtwohill, Ireland). At the end of I60, orlistat and the protease inhibitor Pefabloc (5 mM, Merck, Ireland) were used to inhibit the proteolytic activity of digestive enzymes. A digesta control was included, which was gastrointestinal digestion of digestive fluids and enzymes but with no food added. After I60, this sample was also inactivated with Orlistat and Pefabloc. Before treating cells, digesta samples were diluted 5-fold with HBSS to dilute bile, reduce the effect of inhibitors on Caco-2 brush border enzymes, and reduce osmolality to physiological values [30]. Samples were stored at −20 °C following snap-freezing. Experiments were performed in biological triplicates.

### 2.3. Protein Digestibility

#### 2.3.1. Free Amino Acid Analysis

Free amino acid (AA) analysis was carried out at G0, G60, and I60. Samples were diluted (1:1) with 24% (*w*/*v*) TCA and incubated at room temperature for 10 min prior to centrifugation for 10 min at 14,000 rpm. Supernatants were collected and filtered using 0.45 µm filters, and free AAs were quantified using a Jeol JLC-500/V AA analyzer (Jeol Ltd., Welwyn Garden City, UK) fitted with a Jeol Na^+^ high-performance cation exchange column. Tryptophan is not accurately quantified using this method; therefore, it was not included in the free AA dataset. 

#### 2.3.2. Molecular Weight Distribution of Peptides

Size exclusion (SE) high-performance liquid chromatography (HPLC) was performed on the digesta samples using a TSK G2000SW (300 × 7.5 mm, Tosoh Bioscience LLC, Tosu Hass, Japan) in series, fitted to a Waters Alliance 2695 separation module (Waters Corporation, Milford, MA, USA). The eluent used was 30% (*v*/*v*) acetonitrile (HPLC grade, Merck, Carrigtwohill, Ireland) containing 0.1% trifluoroacetic acid (Merck, Ireland). It was run using a flow rate of 1 mL/min and was continually monitored at 214 nm using a Waters 2487 dual wavelength detector. Filtered (0.45 µm) IMF samples at G0, G60, and I60 time points of digestion (20 µL) were injected into the column at 0.25% protein concentration. Chromatographic data were collected and analyzed using the Empower data handling software package (Waters Corporation, USA). A molecular weight calibration curve, prepared from retention times of standard proteins and peptides, was used to calculate peptide size distribution [31]. Data are presented as the area under the HPLC curve for each range of protein/peptide sizes, <0.4 kDa, 0.4–1 kDa, 1–7 kDa, 7–11 kDa, and >27 kDa molecular weight, calculated as a percentage of the total area under the curve for each sample.

#### 2.3.3. Proteomics

Full details of the proteomics protocol used in this study can be found in Appendix A. Briefly, samples obtained after gastric and intestinal digestion of HC-IMF, LC-IMF, and HBM were lyophilized and dissolved in 1 mL of milli-Q water alongside apical and basolateral supernatants after absorption by Caco-2/HT-29MTX monolayers. An aliquot (50 µL) was desalted by spin column C18 (Pierce Biotechnology, Rockford, IL, USA), and the eluate was dried by speed-vac and then reconstituted in 100 µL formic acid (0.1%). Desalted samples (5 µL) were analyzed by liquid chromatography–tandem mass spectrometry (LC–MS/MS).

The nano-LC–MS/MS analyses were performed using an Ultimate 3000 nanoflow ultrahigh performance liquid chromatography (Dionex/Thermo Scientific, Sunnyvale, CA, USA) coupled to a Q Exactive Orbitrap mass spectrometer (Thermo Scientific, USA). Spectra were elaborated using the software Xcalibur version 3.1 (Thermo Scientific, USA). Each sample was analyzed in triplicate (technical replicate).

For protein identification in the resulting soluble fractions, raw files of the nano-LC−MS/MS runs were used as the output using the Andromeda search engine of the open-source MaxQuant bioinformatics suite (version 2.0.3.0) against the Bos taurus milk protein database (αs1-casein, αs2-casein, β-casein, κ-casein, β-lactoglobulin, α-lactalbumin) (UniProt, Switzerland) for IMF. For HBM, searches were taxonomically restricted to the human databases downloaded from UniProtKB (2022 version, www.uniprot.org). Peptide spectrum matches were filtered using the target decoy database approach with an e value of 0.01 peptide-level false discovery rate, corresponding to a 99% confidence score. Technical replicates were merged, while biological replicates were each considered as a single experiment. The proteomics data have been deposited to the ProteomeXchange Consortium [32] via the PRIDE partner repository with the dataset identifier PXD043255. 

### 2.4. Free Fatty Acid Analysis

Analysis of free fatty acids (FAs) in G0, G60, and I60 digests and basolateral samples was carried out by MS-Omics (Vedbæk, Denmark) as follows: Samples were derivatized with methyl chloroformate using a slightly modified version of the protocol described by Smart et al. [33]. All samples were analyzed in a randomized order. Analysis was performed using gas chromatography (7890B, Agilent, Santa Clara, CA, USA) coupled with a quadruple mass spectrometry detector (5977B, Agilent, USA). The system was controlled by ChemStation (Agilent, USA). Raw data were converted to netCDF format using ChemStation (Agilent, USA) before the data were imported and processed in Matlab R2018b (MathWorks, Inc., Natick, MA, USA) using the PARADISe 6.0.1 (Chemometrics Research, Copenhagen, Denmark) software described by Johnsen et al. [34].

### 2.5. Mineral Analysis 

Mg and Ca standards (0.2–2 μg/mL) were prepared from commercially available solutions. Lanthanum chloride solution was added to samples for Ca analysis. Digested samples were diluted between 1:4 and 1:10 to bring concentrations within the standard curve range. Liquid samples (1 mL) were filtered through a 0.45 μm syringe filter, dried in a crucible, and pre-ashed by charring over a Bunsen burner. Samples were ashed in a muffle furnace (500 °C) and redissolved in HCl (25%) with a few drops of concentrated nitric acid. Atomic absorption spectrophotometry (Varian SpectrAA-600, Palo Alto, CA, USA) was used to analyze mineral content.

### 2.6. Analysis of Intestinal Health

#### 2.6.1. Intestinal Barrier Cell Culture Model

The epithelial colorectal adenocarcinoma cell line, Caco-2 (ATCC, passage 25–40), and the human goblet colon adenocarcinoma cell line HT-29MTX (ATCC, MTX modified, passages 67–78) were cultured as previously described in Dulbecco’s Modified Eagle’s Medium (DMEM) with 10% fetal bovine serum (FBS), 100 U/mL penicillin, and 100 μg/mL streptomycin [35]. Caco-2 and HT-29MTX were seeded at a ratio of 9:1 (6 × 10^4^ cells/well) to the apical compartment of 12-well transwell plates (area of well 1.12 cm^2^). Co-cultures were left to differentiate for 21 days, and media (0.5 mL on the apical and 1.5 mL on the basolateral side) was replaced every 2–3 days. The integrity of the monolayer was tested once a week by transepithelial electrical resistance (TEER) using Millicell-ERS (Merck, Ireland).

Twenty-one-day-old monolayers that had reached a TEER value of ≥500 Ω × cm^2^ were used for the experiment. Prior to exposure, differentiated monolayers were washed twice with Hank’s Balanced Salt Solution (HBSS, Merck, Ireland) and then incubated in fresh HBSS for 30 min at 37 °C and 5% CO_2_. Digested IMF samples (I60) diluted 5-fold in HBSS were filter-sterilized (0.45 μm). The concentration added to the monolayers was 5 mg of starting IMF/cm^2^, which equated to a final protein concentration of approximately 500 µg/cm^2^. Monolayers were incubated with I60 samples for 4 h. Apical and basolateral supernatants were collected and stored at −20 °C prior to analysis. To collect cell lysates, 0.25 mL of ice-cold lysis RIPA buffer containing protease/phosphatase inhibitors (ThermoFisher, Ireland) was added to the monolayers for 5 min. Cell lysates were then collected and stored at −20 °C prior to analysis. TEER measurements were performed before and after washing and after 4 h treatment. 

#### 2.6.2. Cytotoxicity Assays

For cell viability high-throughput assays, Caco-2 and HT-29MTX cells were seeded at a ratio of 9:1 (5 × 10^4^ cells/well) in 96 well plates. After 24 h, media was aspirated, and cells were washed with HBSS, followed by a 4-h incubation with IMF I60 samples (0–200 µg protein/mL) or HBM I60 in the same dilution. After 4 h, 20 μL of CellTiter One solution (Promega, Southampton, UK) was added to wells for a further 1 h. A Synergy plate reader (BioTek, Burlington, VT, USA) was used to measure absorbance at 490 nm in each well, and viability was calculated as a percentage of control (HBSS alone, assigned 100%). Assays were performed on two different days with three technical replicates. 

#### 2.6.3. Immunomodulation Analysis 

Human monocytic THP-1 cells (ATCC, passage number 5–10) were cultured in 25 cm^2^ and 75 cm^2^ flasks in RPMI-1640 ATCC-formulated media (Cat. 30-2001), supplemented with 10% heat-inactivated FBS and 2-mercaptoethanol to a final concentration of 0.05 mM, and incubated at 5% CO_2_ and 37 °C. Cultures were maintained by the addition of fresh growth medium every 2–3 days, with cell counting every second day to ensure optimal cell density (1 × 10^5^ to 1 × 10^6^ cells/mL). Cells were seeded in 24-well plates at a density of 2 × 10^5^ cells/mL and treated with phorbol 12-myristate 13-acetate (5 ng/mL final concentration) in the same media to induce differentiation. THP-1 cells were allowed to differentiate for 2–3 days before the differentiation media was changed back to the standard growth medium for 24 h.

For cell exposure, I60 samples were prepared to a final concentration of 10 mg of initial IMF powder per mL or HBM in the same dilution in a growth medium and filter sterilized. Wells were washed twice with phosphate-buffered saline (Sigma, Arklow, Ireland), and 0.5 mL of each treatment was applied for 4 h. Controls included a vehicle (growth medium only), a positive control (10 ng/mL of filter-sterilized lipopolysaccharide, L5418, Sigma, Ireland), and a negative control (blank digesta, no food control) prepared with the same dilution as each sample. After 4 h, cell supernatants were collected in labeled tubes and stored at −20 °C until biomarker quantification, performed with Milliplex Map Kit (HSTCMAG-28SK-06, Merck, Ireland) and MagPix fluorescent detection system (Luminex, Austin, TX, USA) according to the manufacturer’s instructions. A five-parameter logistic curve fitting method was used in the following ranges for each biomarker: IFN-γ, 0.61–2500 pg/mL; IL-1β, 0.49–2000 pg/mL; IL-6, 0.18–750 pg/mL; IL-8, 0.31–1250 pg/mL; IL-10, 1.46–6000 pg/mL; and TNF-α, 0.45–1750 pg/mL. Experiments were carried out in triplicates, with each replicate of digesta assayed on a different experimental day. 

#### 2.6.4. Satiety Analysis

Analysis of promoted satiety was performed as previously described [36]. Murine enteroendocrine cells STC-1 (passage number 22–28) were cultured at 0.5 × 10^6^ cells/well in 12-well plates in DMEM containing 4.5 g/L of glucose and L-glutamine, supplemented with 10% FBS, 100 U/mL penicillin, and 100 μg/mL streptomycin for 18 h. Cells were washed once with modified Krebs–Ringer buffer (containing 11 mM glucose and 10 mM HEPES without bicarbonate) and then pre-incubated in the same buffer for 1 h to acclimatize. Buffer was then aspirated and replaced with 1 mL of filter-sterilized (0.45 μm) I60 samples (equivalent to 10 mg initial IMF powder/mL or HBM in the same dilution), prepared in modified Krebs–Ringer buffer. Following the 4 h incubation period at 37 °C, 10 μL of 100× Halt Protease and Phosphatase Inhibitor (Thermo Fisher, Ireland) was used to inactivate endogenous dipeptidyl peptidase IV activity. Cell supernatants were collected in 1.5 mL tubes and centrifuged at 900× *g* and 4 °C for 5 min to remove cellular debris. Supernatants were stored at −80 °C prior to quantification, which was performed using a Milliplex Map Kit (HMHEMAG-34K, Merck, Ireland) and MagPix fluorescent detection system (Luminex, Austin, TX, USA) according to the manufacturer’s instructions. The Milliplex assay allowed for the detection of active glucagon-like peptide (GLP-1) levels in the range of 41–30,000 pg/mL, using a five-parameter logistic curve fitting method to quantify samples. Assays were performed with three experimental replicates on different days, with 2–3 technical replicates for each treatment on each day. MagPix quantification of GLP-1 hormone was performed in technical duplicates.

#### 2.6.5. Tight Junction (TJ) Protein Analysis

Western blot analysis of TJ proteins was performed to quantify the effects of digested IMF and HBM samples on the health of the intestinal barrier using Caco-2 and HT-29MTX co-culture monolayers, as previously described [37]. Monolayer cell lysates were defrosted on ice and sonicated twice for 2 min on ice to break cell walls. Total protein concentration was then determined by BCA assay (Pierce™ BCA Protein Assay Kit, Thermo Fisher, Ireland). Cell lysates (25 μg protein) were loaded on denaturing 4–12% w/v polyacrylamide gels (Thermo Fisher, Ireland) and separated using SDS-PAGE. Proteins were transferred to a polyvinylidene difluoride membrane overnight using the wet method. Membranes were blocked overnight with 5% bovine serum albumin in Tris-buffered saline-Tween 20 (TBST) and then incubated for 1 h at 4 °C with either anti-occludin (1:10,000), anti-claudin-1 (1:5000), or anti-β-actin (housekeeping) (1:5000) antibody (Invitrogen, Bio-Sciences, Ireland) diluted in 2% bovine serum albumin in TBST. The membrane was washed three times for 10 min with wash buffer and then incubated with the appropriate secondary (HRP-conjugated) antibody for 1 h before the final washing steps. Each membrane was exposed to the chemiluminescent substrate, SuperSignal™ West Pico PLUS (Thermo Fisher, Ireland), prior to imaging. Images of immunoreactive proteins were collected with the luminescent image analyzer LAS-3000 (Fujifilm, Dublin, Ireland). Image analysis and densitometry scans were performed using Image Lab 3.0 software (Bio-Lab Laboratories, Ballina, Ireland). Western Blot analysis was performed on three different experimental days with pooled lysate samples from 4 to 5 wells of a transwell plate for each digested replicate. Densitometry analyses were performed on triplicate values.

#### 2.6.6. Evaluation of Gut Barrier Integrity with *C. elegans*

The effects of LC-IMF and HC-IMF I60 on gut barrier protection were studied in the *Caenorhabditis elegans* (*C. elegans*) nematode model [38] under two different experimental conditions to determine the preventive and recovery effects of both formulations, using methotrexate (MTX 0.5 μg/mL, Sigma, St. Louise, MO, USA) to induce leaky gut (Figure 1). *C. elegans* N2 wild-type strain was obtained from the Caenorhabditis Genetics Center at the University of Minnesota and maintained at 20 °C on Nematode Growth Medium (NGM) plates with *Escherichia coli* strain OP50 as standard diet for nematodes. Worms were synchronized by isolating eggs from gravid adults at 20 °C.

In the ‘preventive model’, I60 samples were added on the surface of the NGM plates at a dose of 10 μL/mL (equivalent to 0.5 mg initial IMF powder). *C. elegans* were incubated on the NGM plates from the egg stage until the L4 stage at 20 °C. A leaky gut was then induced by transferring L4 to fresh NGM plates in the presence of food (*E. coli* OP50) without I60 samples but with MTX (0.5 μg/mL) for 24 h. Evaluation of the integrity of the nematode’s intestinal barrier was performed by Nile red staining (0.05 μg/mL, Sigma, USA). Staining was quantified using a Nikon SMZ18 fluorescence stereomicroscope (Amstelveen, The Netherlands) equipped with NIS-ELEMENT image software (Nikon, Amstelveen, The Netherlands). Results for the ‘preventive’ assays are shown in terms of the percentage of fluorescence in each IMF treatment compared to the control MTX-challenged population.

For the ‘recovery model’, worms were transferred to NGM plates in the presence of food and exposed to MTX over 24 h to induce damage, stained with Nile red, and then fluorescent staining was visualized to provide data for time point ‘0 h’ after intestinal damage. Worms were then transferred to fresh NGM plates supplemented with IMF samples at a 10 μL/mL dose (equivalent to 0.5 mg initial IMF powder) and then imaged again at 8 h and 24 h time points. ‘Recovery model’ results were shown as the percentage of fluorescence in each treatment compared to the MTX-challenged population. In both preventative and recovery experiments, analysis was performed in duplicate with 60 worms per treatment group. Controls for both models were as follows: negative control (IMF in NGM), positive control (MTX in NGM), and a condition without damage (NGM alone). 

### 2.7. Statistical Analysis

Technical replicates for each analysis are described under their corresponding section header within the Section 2. All analyses were performed at least in duplicate. Statistical analysis of the proteomic data was performed with Perseus 1.6.15.0 (MaxQuant, Max Planck Institute of Biochemistry, Planegg, Germany). Proteins were identified by site reverse, and potential contaminants were filtered prior to analysis. Only peptides identified with at least two technical replicates were retained, while entries with missing values in two out of three replicates were filtered out. In the intestinal gut barrier integrity assay, a parametric one-way analysis of variance with a Tukey post hoc test was applied to compare fluorescence intensity for each condition and the positive control. Statistical analyses were performed in GraphPad Prism 9 package software (GraphPad Software, San Diego, CA, USA). Data are expressed as mean ± standard deviation (SD). A *p*-value <0.05 was considered statistically significant. 

## 3. Results and Discussion

Two IMFs containing different concentrations of cream (HC-IMF and LC-IMF) were formulated at a pilot plant scale and subjected to an *in vitro* static infant gastrointestinal digestion along with HBM. Thereafter, cell-based and *in vitro* assays were used to study the digestibility and bioavailability of key nutrients and immunomodulatory activities. A *C. elegans* leaky gut model was established to investigate the prevention and recovery of intestinal damage with test IMFs. 

### 3.1. Digestion and Bioavailability

#### 3.1.1. Protein Digestion: Peptide Size Distribution by HPLC

Analysis of peptide size distribution by HPLC demonstrated similar profiles of protein degradation and peptide release between LC-IMF and HC-IMF throughout each stage of digestion (Figure 2A). There was a higher percentage of larger proteins (>27 kDa) observed in HBM after the gastric and intestinal phases of digestion than with both IMFs and more very small peptides (<0.4 kDa) in HBM than LC-IMF or HC-IMF at the end of intestinal digestion (Figure 2A).

#### 3.1.2. Peptide Release and Bioavailability 

HBM had a greater diversity (measured by the number of unique identified peptides) of released peptides in the gastric and intestinal phases of digestion and apical supernatants compared to LC-IMF or HC-IMF (*p* < 0.05) (Figure 2B). This result is not surprising considering that HBM contains active endogenous enzymes, which may have partially proteolyzed peptides prior to digestion, whereas heat treatment during IMF processing destroys endogenous enzyme activity [11,39]. HC-IMF had a higher number of unique peptides released in G60 and I60 compared to LC-IMF, though there were no significant differences in the number of peptides in apical and basolateral supernatants (Figure 2B). All milk proteins, with the exception of α-lactalbumin (α-LA), were effectively hydrolyzed and released peptides during the G60 phase of digestion (Figure 2C). Analysis of the relative abundance of peptides in the soluble fraction of simulated digestion indicated that the most abundantly hydrolyzed G60 proteins in LC-IMF were β-casein and αs1-casein, followed to a lesser extent by β-lactoglobulin (β-LG) and κ-casein. Conversely, in HC-MF, the most abundantly hydrolyzed proteins during G60 were αs1-casein, followed by κ-casein, β-LG, β-casein, and αs2-casein. Overall, throughout digestion, more individual unique peptides were identified in HC-IMF, including 62 from α-LA and 381 from β-LG vs. 31 from α-LA and 117 from β-LG in LC-IMF (Appendix A). Dairy cream is known to contain whey proteins, which can explain the higher abundance of α-LA- and β-LG-derived peptides throughout the digestion of HC-IMF vs. LC-IMF. 

Appendix A reports the protein sequence coverage value that represents the number of amino acids in a specific protein sequence found in LC–MS/MS analysis. In HC-IMF, the percentage of the protein sequence covered by identified peptides was 48.6% for α-LA, 90.5% for αs2-casein/casocidin-1, and 82% for β-LG. These values were lower in LC-IMF, where the sequence coverage was 45.8%, 61.3%, and 68.5% for α-LA, αs2-casein/casocidin-1, and β-LG, respectively (Appendix A). Based on the number and intensity of sequenced peptides in the soluble fraction, it can be hypothesized that HC-IMF proteins showed an increased susceptibility to gastric digestion compared to LC-IMF proteins. In all milk samples, the number of unique peptides decreased in the intestinal and apical phases, with a low concentration of peptides detected in the basolateral compartment (the absorbed fraction).

In HBM, the most predominantly adsorbed peptide was the β-casein peptide FDPQIPK, which is recognized for its potential immunomodulatory activity (Table 1) [40]. Indeed, a previous analysis of fractionated *in vitro*-digested peptides from HBM indicated that FDPQIPK was present in a fraction that appeared to have both pro- and anti-inflammatory effects on human macrophages [41]. In the basolateral fraction, the profiles of adsorbed peptides of HC-IMF and LC-IMF were quite similar, both consisting of β-casein and β-LG-derived peptides. However, the abundance of some peptides was up to 8-fold higher in HC-IMF (Table 1). It is also interesting to note that both LC-IMF and HC-IMF released the β-casomorphin precursor, VYPFPGPIPN, which crossed the Caco-2/HT-29MTX monolayers. β-casomorphins are thought to have immune-modulating properties in addition to effects on mineral binding, opioid agonism, thrombin inhibition, antioxidant capacity, and reduction in blood pressure via inhibition of angiotensin 1-converting enzyme (ACE) [42,43]. Therefore, their presence in the LC-IMF and HC-IMF bioavailable fractions may confer potential functional benefits in any of these domains. There were also a number of β-LG peptides present only in HC-IMF, namely peptides GLDIQ, DAQSAPL, and IPAVFK. GLDIQ has been associated with hypocholesterolemic and antihypertensive activities [44,45], while DAQSAL and IPAVFK have been reported to have ACE-inhibitory activity and IPAVFK exhibiting antibacterial effects [46,47]. Previously published work described the presence of GLDIQ in Caco-2 basolateral solutions of IMF produced by cascade membrane filtration, while it was notably absent in samples treated with IMF produced by high temperature processing [37]. The presence of these peptides in HC-IMF could indicate the potential bioactivity of HC-IMF, although additional studies are needed to investigate these peptides from a health perspective.

#### 3.1.3. Free Amino Acid (AA) and Fatty Acid (FA) Release and Bioavailability

HC-IMF released significantly more free AAs in the gastric phase (G60) compared to LC-IMF (Figure 3A; *p* < 0.05), but no significant differences were observed between HC-IMF and LC-IMF at other stages of digestion or in the concentrations of basolateral free AAs (Figure 3B). HBM had significantly more free AAs prior to digestion and after the gastric phase compared to either IMF (Figure 3A; *p* < 0.05). In the intestinal phase, AA release slowed down in HBM; however, there was no significant difference in the level of absorbed AAs compared to either IMF (Figure 3B). Concentrations of individually released free AAs at each stage of digestion are presented in Appendix A. 

HC-IMF released the largest concentration of free FAs in both the gastric and intestinal phases of digestion, followed by LC-IMF and then HBM (*p* < 0.05; Figure 3C). The same pattern of FA release was noted for absorbed free FAs (Figure 3D). The concentration of free FA in HBM prior to digestion was higher than in IMFs (Figure 3C), which could be attributed to the activity of endogenous lipases in HBM. Although these initial data require further validation, the apparent increase in FA release from HC-IMF digestion could indicate the potential for enhanced lipid absorption when using a high cream supplement in IMF, which is supported by the observation of higher simulated FA absorption with HC-IMF. With regard to specific FAs at I60, there were higher concentrations of C16:1, C16:0, C18:2, C22:6, and C20:4 released by HC-IMF compared to LC-IMF (Figure 3C). Our observation of enhanced digestibility of HC-IMF compared with LC-IMF could be attributable to the higher concentration of C:16 β-palmitate in HC-IMF compared with LC-IMF. It should be noted that cream supplementation of IMF would provide multiple FAs, as well as PA and additional nutrients, which may also have contributed to this observation. For example, the increase in C18:2, known as linoleic acid, from HC-IMF versus LC-IMF was statistically significant (*p* < 0.05) and may have relevance for the functionality of HC-IMF. Linoleic acid is an essential FA found in HBM, where it acts as a precursor for arachidonic acid and plays a role in inflammatory responses, immune function, growth, and neurodevelopment [3]. It is important to note that absorbed free FAs are inclined towards shorter chains versus the I60 pool. HC-IMF also had significantly more bioavailable caproic acid (C6:0) compared to LC-IMF and significantly more caproic and capric acid (C10:0) compared to HBM. Both caproic acid and capric acid are hypothesized to have antiviral and antifungal properties [48,49], while capric acid may play a role in the maintenance of healthy cholesterol levels [50,51]. Though these data require further validation, they could highlight the potential immune-modulating benefits of IMF enrichment with cream compared to HBM.

#### 3.1.4. Mineral Release and Bioavailability

The effect of cream concentration in IMF on mineral absorption was studied and compared to HBM. Ca and Mg concentrations were similar between LC-IMF and HC-IMF throughout digestion and absorption (Figure 4). Although Ca and Mg concentrations were significantly lower in HBM compared to IMF, these minerals demonstrated higher bioavailability from HBM. This resulted in no significant difference in concentrations of Ca in basolateral supernatant after cell monolayer exposure to HBM and either LC-IMF or HC-IMF I60 (Figure 4A, *p* > 0.05), suggesting that both IMFs supported calcium absorption. The bioavailable concentration of Mg, however, was significantly greater with HBM compared to IMFs (Figure 4B; *p* < 0.05).

### 3.2. Functionality of HBM, LC-IMF and HC-IMF

#### 3.2.1. Intestinal Health

None of the tested samples at concentrations up to 200 mg/mL reduced viability below 80% compared to the HBSS control. There was no difference in the expression of claudin-1 and occludin in 21-day-old Caco-2/HT29-MTX co-cultures after a 4 h incubation with HBM, LC-IMF, or HC-IMF digesta (Figure 5A,B). TEER values were elevated after incubation with HBM, LC-IMF, or HC-IMF digesta vs. HBSS control (Figure 5C). Treatment with no-food control resulted in a TEER value 1.46-fold that of the control TEER value, which can be an effect of bioactive small peptides generated from enzyme autolysis. All tested samples significantly increased TEER of monolayers compared to HBSS vehicle control; however, no statistically significant differences were noted between samples. Taken together, these data indicate IMF digesta, regardless of cream content, has similar effects to HBM digesta on the integrity of the *in vitro* intestinal barrier model. 

#### 3.2.2. Satiety

To investigate the effect of cream as a source of lipids in IMF on its hormone-mediated satiety effect, levels of the satiety hormone GLP-1 were tested after exposure of enteroendocrine STC-1 cells to digested IMFs and HBM. Secretion of GLP-1 was not significantly affected by exposure to HBM, LC-IMF, or HC-IMF I60 samples compared to vehicle control (Krebs buffer alone) after 4 h incubation, suggesting that the effects of IMF and HBM on satiety were similar (Figure 5D). The no-food digestive control resulted in 23 pM of active GLP-1 secretion, which was not different from the Krebs control and test samples (*p* > 0.05).

#### 3.2.3. Immunity/Inflammation

To track the immune response to IMF digesta, THP-1 cells were treated with I60 samples. THP-1 cells were selected as a model to study immunity effects, as they can be differentiated into macrophages and produce immune biomarkers at the basal level without stimulation. Exposure of THP-1 cells to HC-IMF, LC-IMF, or HBM digesta resulted in comparable levels of secreted IFN-γ, IL-8, and TNF-α; however, IL-8 and TNF-α were significantly lower after exposure to IMF and HBM digesta compared to vehicle control (*p* < 0.05; Figure 6). IL-10 and IL-1β concentrations were significantly higher after HBM I60 exposure compared to LC-IMF I60 exposure (*p* < 0.05), whereas expression of both these markers was similar in cells treated with the I60 samples of either HC-IMF or HBM (Figure 6B,C). Our observations are in line with previous findings, whereby exposure of Caco-2 cells to HBM lipids resulted in elevated expression of the anti-inflammatory cytokine IL-10, indicating the potential of HBM to modulate cytokine expression in the newborn gut [52]. Similarly, Hahn et al. [53] reported that treatment of THP-1 cells with HBM resulted in rapid upregulation of IL-1β mRNA compared to untreated cells. Immune-modulating properties of HC-IMF appear more in line with HBM than LC-IMF, which could be an important consideration in the optimization of IMF. The rate of infant infection is generally lower in infants who are exclusively breastfed due to immune-modulating attributes, including acquired maternal immunity, of breast milk [54,55,56,57]. Thus, supplementing IMF with a component that has similar immune-modulating properties to HBM, such as cream, could be advantageous.

#### 3.2.4. Intestinal Health Effect in *C. elegans*

Even though the *in vitro* experimental setup allowed us to compare IMFs and HBM at a mechanistic level, cell models have limited translation to longer feeding effects. To track the prevention and recovery of gut barrier damage *in vivo* and further our comparative investigations of IMFs, *C. elegans* nematodes were fed with I60 digests of IMFs. Due to the sample volume required, only IMFs were assessed. Nematodes were treated with MTX, and gut barrier damage was confirmed by Nile red staining. Nematodes with intestinal permeability show increased fluorescence intensity due to the infiltration of the dye into the internal cavity of the worm. In the ‘prevention model’, LC-IMF and HC-IMF both significantly improved intestinal barrier integrity and prevented dye leakage from the intestinal cavity in a similar fashion, compared to the positive control—MTX alone (Figure 7A). In the recovery model, feeding worms with HC-IMF resulted in significantly lower intestinal leakage at the 8 h time point after intestinal damage with MTX vs. LC-IMF or positive control (Figure 7B). This was demonstrated by the significant reduction in Nile red staining (*p* < 0.05), suggesting quicker recovery from intestinal damage. By 24 h, there was no significant difference in the intestinal barrier permeability across all treatments (Figure 7B).

The observation that exposure to HC-IMF may aid in more rapid recovery from intestinal damage, compared to LC-IMF, may also have relevance when considering the overall functionality of future formulations. These results may be attributed to the higher proportion of MFG/MFGM in the HC-IMF than in the LC-IMF. Previous studies have shown that MFGM demonstrates anti-inflammatory properties, can improve intestinal health, and can prevent damage [9]. Taken together with our observation that the expression of TJ proteins and monolayer integrity was similar following exposure to IMF and HBM digests, our data suggest that high cream supplementation supports gut health with additional gut-protective/repair benefits. These findings may be due to the improved FA profile of HC-IMF and similarity in overall composition compared with HBM as a result of adding cream.

## 4. Concluding Remarks

Though there are already some commercially available IMF-containing creams, this study is, to the best of our knowledge, the first published account of the effects of cream supplementation on the nutritional value and functionality of IMF. By making our data available in this way, we hope to be able to enrich the understanding of the importance of fatty acid composition in the development of IMF in order to drive the quality of future IMFs currently in development. Future studies should include a non-cream control, although this will introduce compositional variability in minor components. The infant gut barrier model can be applied in future *in vitro* studies for the most state-of-the-art and physiologically relevant outcomes [35].

A particular strength of this study is that the formulations used were tightly controlled to ensure that they possessed an identical nutritional profile and process history and differed only in the amount of cream contributing to this profile. Similarly, HBM was taken from a pooled milk supply rather than a single milk source to generate a more representative sample to study. A mixture of advanced *in vitro* cell models and *in vivo* experiments were used to assess the properties of each formulation, as well as their functional effects, to provide a more complete analysis of the potential effects of cream supplementation in IMF on infant nutrition. Thus, our preliminary findings warrant further validation in more advanced systems, such as the piglet, and ultimately, clinical trials investigating the efficacy and safety of infant-grade dairy cream in IMF.

In summary, HC-IMF and LC-IMF had comparable benefits to HBM in a number of the paradigms tested in this study. Although minimal differences in the absorption of free FAs, immune-modulating properties, and intestinal recovery were observed, it could be concluded that LC-IMF and HC-IMF performed similarly. These initial observations suggest that using a cream with IMF provides added nutritional and functional benefits.

## Figures and Tables

**Figure 1 nutrients-16-03065-f001:**
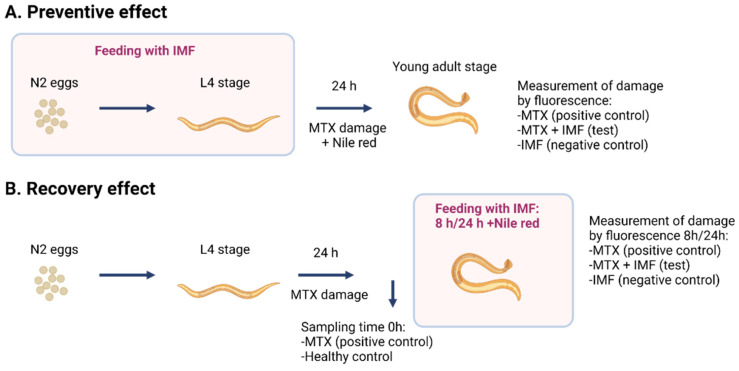
Schematic representation of the *C. elegans* model of intestinal damage: preventive (**A**) and recovery (**B**) experimental setup for intestinal damage in the *C. elegans* model. Concentrations used were as follows: IMF (10 μL/mL); MTX (0.5 μg/mL); Nile red (0.05 μg/mL). Sixty worms were used per treatment, and experiments were performed in duplicates. *C. elegans*, *Caenorhabditis elegans*; MTX, methotrexate.

**Figure 2 nutrients-16-03065-f002:**
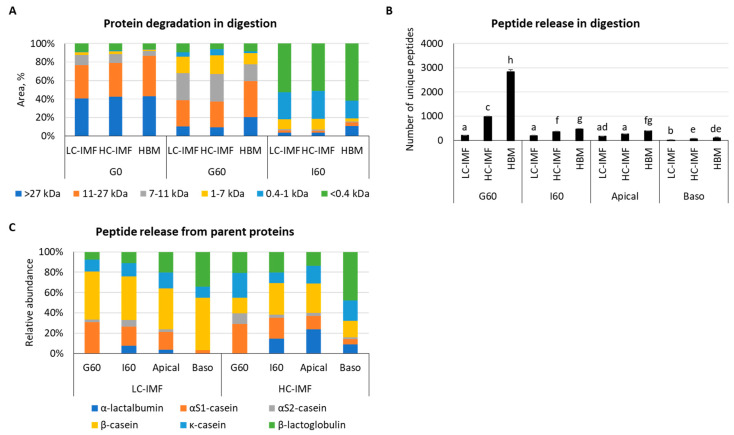
Peptide release and absorption from HBM, HC-IMF, and LC-IMF in infant gastrointestinal digestion. Molecular weight distribution of proteins and peptides by SE-HPLC prior to the digestion (G0), after the gastric phase (G60), and after the intestinal phase (I60) (**A**). Number of released individual peptides in stages of simulated infant digestion and absorption by LC–MS/MS in LC-IMF, HC-IMF, and HBM. Different letters indicate significant differences in number of individual unique peptides, *p* < 0.05 (**B**). Relative abundance of released individual peptides from different parental proteins in stages of simulated infant digestion and absorption by LC–MS/MS in LC-IMF and HC-IMF (**C**). N = 3. HC-IMF, high-cream IMF; LC-IMF, low-cream IMF.

**Figure 3 nutrients-16-03065-f003:**
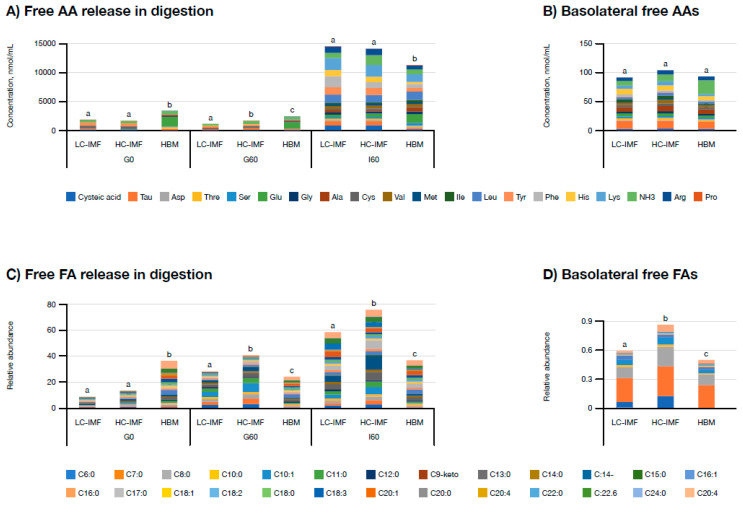
Free AAs and free FAs release and absorption from HBM, LC-IMF, and HC-IMF. Profile of free AAs release (**A**,**B**) and free FAs release (**C**,**D**) in simulated gastrointestinal digesta and after the simulated absorption with 21-day-old Caco-2/HT-29MTX monolayers over 4 h incubation with I60 samples at 500 μg protein/cm^2^ in HBSS buffer. N = 3. Different letters within a time point indicate significant differences in total AAs or FAs concentrations, *p* < 0.05. Raw data on free AA release are provided in Appendix A.

**Figure 4 nutrients-16-03065-f004:**
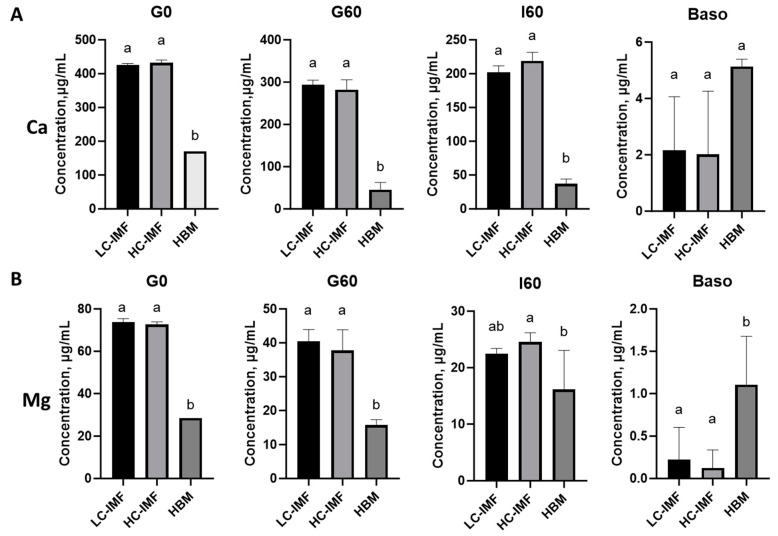
Mineral release and absorption in HBM, LC-IMF, and HC-IMF. The concentration of Ca (**A**) and Mg (**B**) in IMF and HBM prior to the digestion (G0), after the gastric phase (G60), after the intestinal phase (I60), and after absorption across 21-day-old Caco-2/HT29-MTX monolayers at 500 μg protein/cm^2^. Values are represented as mean with SD as error bars, N = 3. Different letters at each time point indicate significant differences (*p* < 0.05). HC-IMF, high-cream IMF; LC-IMF, low-cream IMF; SD, standard deviation.

**Figure 5 nutrients-16-03065-f005:**
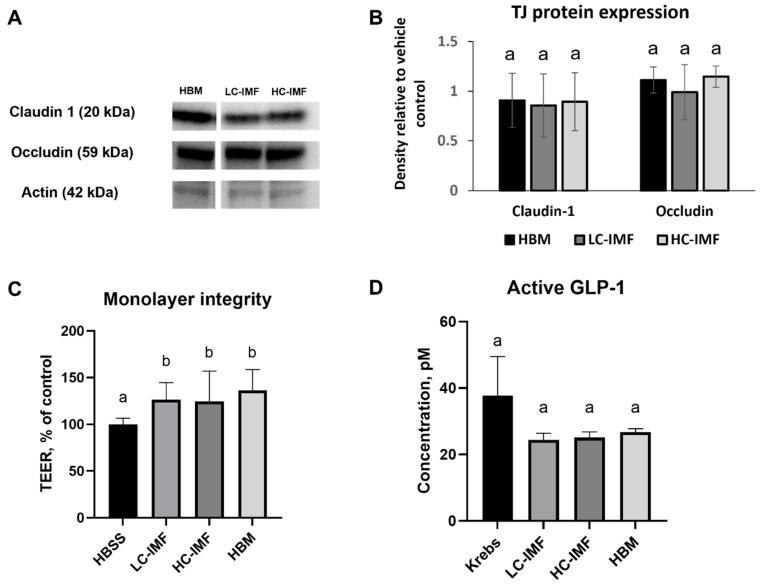
Functional effects. Tight junction (TJ) protein expression in 21-day differentiated monolayers after 4 h incubation with IMF I60 samples (at 500 μg protein/cm^2^ in HBSS) (**A**), and quantification of the bands by densitometry (**B**), N = 3. Intestinal barrier monolayer integrity by TEER in 21-day differentiated monolayers after 4-h incubation with IMF samples (**C**). Monolayers in the HBSS buffer had an average TEER of 664 ± 72 Ω×cm^2^ and were assigned a value of 100%, N = 18. Release of satiety hormone GLP-1 (active) over 4 h incubation with 0.5 × 10^6^ cells/well in 12-well plates STC-1 cells treated with 500 μg protein/cm^2^ of IMF I60 digesta in Krebs buffer (**D**), N = 9. Different letters for each treatment indicate significant differences (*p* < 0.05). HC-IMF, high-cream IMF; LC-IMF, low-cream IMF.

**Figure 6 nutrients-16-03065-f006:**
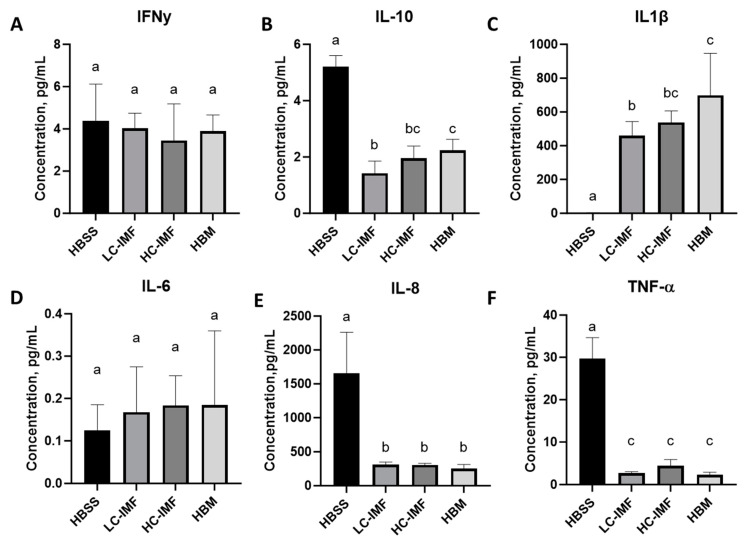
Immunomodulatory properties. THP-1 cells (5 × 10^5^ cells/well in 12 well plates) were differentiated for 3 days and then treated with digested IMF samples (at 500 μg protein/cm^2^ in HBSS) or HBM in the same dilution for 4 h. Concentrations of secreted IFN-ɣ (**A**), IL-10 (**B**), IL1-β (**C**), IL-6 (**D**), IL-8 (**E**), and TNF-α (**F**) are represented as mean with SD as error bar, N = 6. Different letters at each time point indicate significant differences (*p* < 0.05). HC-IMF, high-cream IMF; LC-IMF, low-cream IMF; SD, standard deviation.

**Figure 7 nutrients-16-03065-f007:**
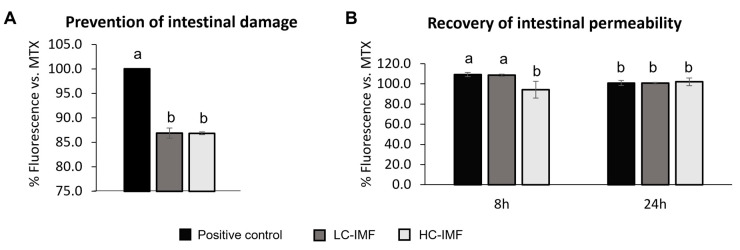
Intestinal damage. Effect of LC-IMF and HC-IMF on the recovery (**A**) and prevention (**B**) of intestinal barrier damage by MTX (positive control) in a *C. elegans* model. Prevention and recovery effects were determined by the intensity of Nile red staining of the body after leakage from the intestinal cavity, which was expressed as a percentage of fluorescence for each treatment group compared to the MTX-treated group (positive control). Concentrations used were as follows: IMF (10 μL/mL); MTX (0.5 μg/mL); Nile red (0.05 μg/mL). Sixty worms were used per treatment, and experiments were completed in duplicate. Values are represented as mean with SD as error bar, N = 3. Values without common letters differ significantly from each other (*p* < 0.05). *C. elegans*, *Caenorhabditis elegans*; HC-IMF, high-cream IMF; LC-IMF, low-cream IMF; MTX, methotrexate; NGM, normal growth medium; SD, standard deviation.

**Table 1 nutrients-16-03065-t001:** Bioavailable peptides, identified by LC–MS/MS in a basolateral solution of Caco-2/HT29-MTX monolayers after 4 h exposure to LC-IMF, HC-IMF, and HBM (N = 3) and their relative abundance.

LC-IMF		HC-IMF		HBM	
Peptide	Intensity, 10^6^	Peptide	Intensity, 10^6^	Peptide	Intensity, 10^6^
(βLg)-VEELKPTPE	16.59	(βLg)-VEELKPTPE	132.31	(βCn)-FDPQIPK	1116.60
(βLg)-YVEELKPTPE	9.82	(βLg)-YVEELKPTPE	42.15	(βCn)-PEIMEVPK_Oxidation (M)	176.72
(βLg)-YVEELKPTPE_Lac(K)	7.04	(βLg)-GLDIQ	40.65	(βCn)-QQVPQPIPQ	74.41
(βLg)-VEELKPTPE_Lac (K)	4.85	(βLg)-EELKPTPE	35.76	(βCn)-DPQIPK	15.85
(βCn)-EMPFPK	31.18	(βLg)-VEELKPTPE_Lac (K)	24.33	(βCn)-PLMQQVPQPI	12.75
(βCn)-HLPLPL	9.62	(βLg)-IPAVFK	13.33	(αS1Cn/Casoxin-D)-VQVP	88.95
(βCn)-QEPVLGPV	7.40	(βLg)-ELKPTPEGDLEIL	10.89	(αLA)-FLDDDITDDI	32.45
(βCn)-IPPLTQTPV	6.61	(βLg)-DAQSAPL	8.32	(multiple *)-RMD	1017.60
(βCn)-MHQPHQPLPPT	5.54	(βCn)-EMPFPK	69.03	(multiple)-EDLSDEAERDE	10.60
(βCn)-NIPPLTQTPV	5.53	(βCn)-EMPFPK_Oxidation (M)	29.22	(multiple)-DLSDEAERDE	9.58
(βCn)-VVPPFLQPEV	4.18	(βCn)-VVPPFLQPE	24.51	(multiple)-NEESTIPR	7.62
(βCn)-VYPFPGPIPN	3.96	(βCn)-HLPLPL	24.03	(multiple)-KHE	5.57
(βCn)-HQPHQPLPPT	3.33	(βCn)-VYPFPGPIPN	10.66	(multiple)-YQL	4.94
(KCn)-NQDKTEIPT	6.07	(βCn)-HQPHQPLPPT	10.01	(multiple)-MKT	4.92
(KCn)-ESPPEINT	4.20	(βCn)-VVPPFLQPEV	8.66	(multiple)-VNEESTIPR	3.81
(αS1Cn)-APSFSDIPNPI	4.84	(KCn)-NQDKTEIPT	78.77	(multiple)-YLH	2.68
(αS1Cn)-HQGLPQ	3.60			(multiple)-YYP	1.03

* Where peptide origin is indicated as “multiple,” parental proteins could be two or more of: S-acyl fatty acid synthase thioesterase (medium chain), Beta-1,4-galactosyltransferase 1, Beta-N-acetylglucosaminyl-glycolipid beta-1,4-galactosyltransferase, Beta-N-acetylglucosaminylglycopeptide beta-1,4-galactosyltransferase, Lactose synthase A protein, N-acetyllactosamine synthase, Processed beta-1,4-galactosyltransferase 1, Clusterin, Inter-alpha-trypsin inhibitor (heavy chain H2), Xanthine dehydrogenase, Xanthine dehydrogenase/oxidase, Xanthine oxidase, Angiopoietin-related protein 4, Cytoplasmic aconitate hydratase, Galectin-3-binding protein, Polymeric immunoglobulin receptor, Kaliocin-1, Lactoferricin-H, Lactoferroxin-A, Lactoferroxin-B, Lactoferroxin-C, Lactotransferrin, Polymeric immunoglobulin receptor (secretory component), Bradykinin, Calreticulin, Kininogen-1, Kininogen-1 (heavy chain), Kininogen-1 (light chain), Low molecular weight growth-promoting factor, Lysyl-bradykinin, Macrophage mannose receptor 1, T-kinin.

## Data Availability

The data presented in this study are available in ProteomeXchange Consortium via the PRIDE partner repository with the dataset identifier PXD043255 or from the corresponding author.

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
