# Peer review of "Infant Milk Formula Enriched in Dairy Cream Brings Its Digestibility Closer to Human Milk and Supports Intestinal Health in Pre-Clinical Studies"

_nutrients, 2024, doi:10.3390/nu16183065_

Round 1

Reviewer 1 Report

Comments and Suggestions for Authors

This study compared the compositions in spray-dried infant formula and donor milk, as well as the gut digestibility and health using human cell line and a roundworm gut model. This study is comprehensive and I love it! The manuscript is well written and data is well presented. However, very minor revision is needed.

I'd suggest to change the term 'human breast milk' or 'breast milk' to 'human milk'. Also, 'infant milk formula' should be 'infant formula milk' or 'infant formula'.

Lane 137: Could it be 'double of level of cream in LC -IMF when compared to HC-IMF'?

Section 2.7: Which statistical tests were used? Parametric or non-parametric tests?

Author Response

Reviewer 1.

This study compared the compositions in spray-dried infant formula and donor milk, as well as the gut digestibility and health using human cell line and a roundworm gut model. This study is comprehensive and I love it! The manuscript is well written and data is well presented. However, very minor revision is needed.

Authors would like to thank reviewers for high evaluation and are happy to incorporate suggestions to the final manuscript.

I'd suggest to change the term 'human breast milk' or 'breast milk' to 'human milk'. Also, 'infant milk formula' should be 'infant formula milk' or 'infant formula'.

Thank you for this comment, authors believed that term 'human breast milk' is easier to abbreviate to HBM to be recognisable across the text. We corrected now to appear HBM throughout the manuscript after the first mentioning. ‘Infant milk formula’ is a common term to be abbreviated to IMF and can support recognition of term for readers. Now corrected to IMF throughout manuscript after the first mentioning.

Lane 137: Could it be 'double of level of cream in LC -IMF when compared to HC-IMF'?

We thank reviewer for finding this typo, corrected.

Section 2.7: Which statistical tests were used? Parametric or non-parametric tests?

We added information in Statistical analysis section to specify that our data were normally distributed and thereby parametric One-way ANOVA was used with Tukey’s post hoc.

Reviewer 2.

Infant milk formula enriched in dairy cream brings its digestibility closer to human milk and supports intestinal health. The digestibility and potential health functions of IMF containing low cream (LC-) or high cream (HC-) was compared with pooled HBM. Bioavailability of key nutrients and immunomodulatory activities were determined via cell-based in-vitro assays. It was reported that the immune-modulating properties of HC-IMF appeared to be more similar to HBM than LC-IMF, as observed by comparable secretion of cytokines IL-10 and IL-1β from THP-1 macrophages. HC-IMF also supported intestinal recovery in C.elegans following distortion, versus LC-IMF. It was concluded that increasing the cream content to 16.5% in IMF may provide added nutritional and functional benefits more aligned with HBM. 

The manuscript is well written and organized, the methods are well described, the results are well explained. 

There are few points needed to be addressed.

The title should contain the term in vitro

Thank you for noting this, we have included “in pre-clinical studies” to specify the type of research performed, as “in vitro” would not cover C.elegans model used.

If possible, add SEM instead of SD

As a standard in pre-clinical research, we provided standard deviations as a way to transparently demonstrate scattering of data and variability.

In the result section, you started the result with description of methods, this needed to be corrected.  You already described the methods, so I suggest you revise the following by adding these lines to the method section or remove them if they were already mentioned in the methods:

L390-395 – Corrected.

L398-400 – Corrected.

L417-420– Corrected.

L491-493 – Corrected.

L555-557 – Corrected.

Reviewer 2 Report

Comments and Suggestions for Authors

Infant milk formula enriched in dairy cream brings its digestibility closer to human milk and supports intestinal health. The digestibility and potential health functions of IMF containing low cream (LC-) or high cream (HC-) was compared with pooled HBM. Bioavailability of key nutrients and immunomodulatory activities were determined via cell-based in-vitro assays. It was reported that the immune-modulating properties of HC-IMF appeared to be more similar to HBM than LC-IMF, as observed by comparable secretion of cytokines IL-10 and IL-1β from THP-1 macrophages. HC-IMF also supported intestinal recovery in C.elegans following distortion, versus LC-IMF. It was concluded that increasing the cream content to 16.5% in IMF may provide added nutritional and functional benefits more aligned with HBM. 

The manuscript is well written and organized, the methods are well described, the results are well explained. 

There are few points needed to be addressed.

The title should contain the term in vitro

If possible, add SEM instead of SD

In the result section, you started the result with description of methods, this needed to be corrected.  You already described the methods, so I suggest you revise the following by adding these lines to the method section or remove them if they were already mentioned in the methods:

L390-395

L398-400

L417-420

L491-493

L555-557.

Author Response

(The authors gave the same response as above.)
